# mRNA-Seq and miRNA-Seq Analyses Provide Insights into the Mechanism of *Pinellia ternata* Bulbil Initiation Induced by Phytohormones

**DOI:** 10.3390/genes14091727

**Published:** 2023-08-29

**Authors:** Wenxin Xu, Haoyu Fan, Xiaomin Pei, Xuejun Hua, Tao Xu, Qiuling He

**Affiliations:** 1College of Life Sciences and Medicine, Zhejiang Sci-Tech University, Hangzhou 310018, China; xuwenxinwenyan@outlook.com (W.X.); zsfhydy@163.com (H.F.); 315532018603@163.com (X.P.); xuejun_h@126.com (X.H.); 2Laboratory of Plant Secondary Metabolism and Regulation of Zhejiang Province, Hangzhou 310018, China

**Keywords:** induction, bulbil, *P. ternata*, microRNA, phytohormone, RNA-seq

## Abstract

*Pinellia ternata* (Thunb.) Breit (abbreviated as *P. ternata*) is a plant with an important medicinal value whose yield is restricted by many factors, such as low reproductive efficiency and continuous cropping obstacles. As an essential breeding material for *P. ternata* growth and production, the bulbils have significant advantages such as a high survival rate and short breeding cycles. However, the location effect, influencing factors, and molecular mechanism of bulbil occurrence and formation have not been fully explored. In this study, exogenously applied phytohormones were used to induce in vitro petiole of *P. ternata* to produce bulbil structure. Transcriptome sequencing of mRNA and miRNA were performed in the induced petiole (TCp) and the induced bulbil (TCb). Gene Ontology (GO) term enrichment and Kyoto Encyclopedia of Genes and Genomes (KEGG) pathway analysis were performed for the identification of key genes and pathways involved in bulbil development. A total of 58,019 differentially expressed genes (DEGs) were identified. The GO and KEGG analysis indicated that DEGs were mainly enriched in plant hormone signal transduction and the starch and sucrose metabolism pathway. The expression profiles of miR167a, miR171a, and miR156a during bulbil induction were verified by qRT-PCR, indicating that these three miRNAs and their target genes may be involved in the process of bulbil induction and play an important role. However, further molecular biological experiments are required to confirm the functions of the identified bulbil development-related miRNAs and targets.

## 1. Introduction

*P. ternate,* the most widely used herb in the *Pinellia* genus, is widely distributed in China, Japan, and Korea [1]. The dried tuber of *P. ternata*, called *Pinellia* rhizome (PR), is a commonly used medicinal plant after processing and is often used to treat phlegm, cough, and morning sickness [2]. However, propagating *P. ternata* is difficult because of viral infection, continuous cropping obstacles, unstable quality, and other factors [3]. As a special vegetative propagule, the bulbil is usually found in the aboveground parts of perennial herbs, such as *Lilium lancifolium* [4], *Dioscorea alata* [5], and *P. ternate* [6].

As a vegetative viviparous organ species, the bulbil has the characteristics of strong vitality, high germination rate, and fast germination cycles [3]. Moreover, the bulbil’s morphology, occurrence site, and number of different species have great variations [7]. The occurrence site of bulbils of *P. ternata* generally involves the petiole near the root, and terminal globules occasionally emerge at the base of three leaves [8]. The anatomical study of the normal development of *P. ternata* bulbils indicated that these developed from the dedifferentiation and redifferentiation of the parenchyma cells between the ventral epidermis of young petioles and the outermost vascular bundles [9]. In vitro petioles of *P. ternata* induced by MS medium containing a plant hormone comprising 0.5 mg/L 6-BA (N-(Phenylmethyl)-9H-purin-6-amine) + 0.2 mg/L NAA (Naphthaleneacetic acid) can obtain a large number of vegetative structures that are similar to bulbils, which can be used in production. They exhibited the same biological characteristics as normal bulbils (such as the cork, capability of rooting, and growth points), and can develop into complete plants [10]. The process of bulbil formation induced by plant hormones can directly complete the structural transformation from explant to bulbil without undergoing the callus stage [11]. This process involves the influence of complex internal and external factors and precise gene expression regulation mechanisms. Evidence suggests that various plant hormones, i.e., 3-Indoleacetic acid (IAA), jasmonic acid (JA), abscisic Acid (ABA), and trans-Zeatin-riboside (ZR), play an important role in the development of bulbils [12,13]. However, exploration at the molecular level still needs to be improved.

The development of bioinformatics and molecular biology supports the study of gene expression in non-model organisms, especially medicinal plants that are relatively scarce but have significant medicinal and research value [14,15]. The bulbil propagation of *P. ternata* has important application value in practical production. Further, despite continuous exploration of influencing factors, key genes, and analytical mechanisms in bulbil formation, these features have thus far failed to be fully elucidated [6,16]. In this study, we used plant hormones to induce bulbil formation and employed RNA-seq and qRT-PCR for the expression of quantitative critical genes in the process of bulbil induction to explore another process of bulbil formation and provide a reference for research on the bulbil initiation and development mechanism.

## 2. Materials and Methods

### 2.1. Plant Materials

The tubers were collected from the planting base in Qingshui, Gansu Province under the permission of the competent authority. The seeds were identified as *P. ternata* tubers by Professor Xu Tao of Zhejiang sci-tech University, and planted in the greenhouse. The soil composition was nutrient soil: vermiculite: perlite = 3:1:1; this was irrigated with 1/5 MS nutrient solution and watered once a week. The temperature setting of the greenhouse where the tubers were cultured was within the range of 25 ± 1 °C with a photoperiod of 12 h. After 20 days of bulb culture, the petioles of the growing *P. ternata* plantlets were cut into 3 cm segments as explants.

After being treated with 70% alcohol (30 s) and 10% sodium hypochlorite solution (10 min), the explants were cultured in MS medium containing 0.5 mg/L 6-BA and 0.2 mg/L NAA. Then, with the time gradient of 3, 6, 9, and 12 days, the tip of the petiole was cut at 0.3 cm for the induction of the bulbil, and at the bottom 0.3 cm for the induction of the petiole (Figure 1A–D). In order to show the completeness of the development of the bulbil, the development and growth of the bulbil at 36 days and 108 days were assessed (Figure 1E,F). Naturally occurring petioles were used as blank controls. mRNA and miRNA sequencing were performed on the inducible petiole and the inducible bulbil on day 6, because the structure of the inducible bulbil could be clearly observed. Five biological replicates of each period, each consisting of 10–15 individual samples, were pooled into two samples prior to sequencing. Three biological replicates were set for each period for qRT-PCR validation. All samples were frozen with liquid nitrogen and then transferred to −80 °C for preservation.

### 2.2. RNA Extraction, RNA Sequencing, and Small RNA Sequencing

Total RNA was extracted from each sample using the Spin Column Plant Total RNA Purification Kit (Sangon Biotech, Shanghai, China) and the RNase-Free DNase Set (Sangon Biotech, China), following the manufacturer’s protocol. The purity and concentration of each RNA sample were analyzed using the Bioanalyzer 2100 and RNA 1000 Nano Lab Chip Kit (Agilent, Santa Clara, CA, USA), with a RIN number > 7.0. Subsequently, the RNA samples were sent to LianChuan Corporation (Hang Zhou, China) for construction of RNA and small RNA (sRNA) sequencing libraries.

### 2.3. Analysis of mRNA-Seq

Firstly, in-house Cutadapt [17] and Perl scripts were used to remove the low-quality, undetermined, and adaptor contamination reads. Then, sequence quality was verified using FastQC (http://www.bioinformatics.babraham.ac.uk/projects/fastqc/, accessed on 20 February 2023). De novo assembly of the transcriptome was performed with Trinity 2.4.0 [18]. All assembled Unigenes were aligned against the Gene Ontology (GO), non-redundant (Nr) protein database, SwissProt, Kyoto Encyclopedia of Genes and Genomes, and eggNOG databases using DIAMOND [19] with a threshold of E value < 0.00001. Salmon [20] was used to determine expression level for Unigenes by calculating TPM [21]. The differentially expressed Unigenes were selected with log2 (fold change) > 1 or log2 (fold change) < −1 and with statistical significance (*p* value < 0.05) using the R package edgeR [22].

### 2.4. Analysis of miRNA-Seq

The small RNA raw data were filtered using an in-house program, ACGT101-miR (LC Sciences, Houston, TX, USA). After removing the adapter dimers, junk, low complexity sequences, common RNA families, and repeats, only unique sequences with length of 18~25 nucleotide were retained and aligned to specific species precursors in miRBase 22.0 to identify known, novel 5p- or 3p- derived miRNA, and novel miRNAs. In the analysis of miRNA expression levels, differential expression of miRNAs based on normalized deep-sequencing counts was analyzed using Fisher exact test, chi-squared 2 × 2 test, chi-squared nXn test, Student’s *t*-test, or ANOVA based on the experiment design. The significance threshold was set to 0.01 and 0.05 in each test. The computational target prediction algorithms (PsRobot, v1.2) were used to identify miRNA binding sites to predict the genes targeted by the most abundant miRNAs. The GO terms and KEGG pathway of the most abundant miRNA targets were also annotated.

### 2.5. qRT-PCR Validation

The QuantStudio 3 real-time PCR system (Thermo Scientific^TM^ EP0733, Santa Clara, CA, USA) was used to validate the accuracy of RNA-seq and the expression profile of key genes. The total RNA extraction and purification methods were consistent with those described above. Due to differences in miRNAs and genes, reverse transcription uses different processes. First, 250 ng RNA from each sample was used to synthesize single-stranded miRNA cDNA using a miRNA 1st Strand cDNA Synthesis Kit (by stem-loop) (Vazyme Biotech Co., Ltd., Nanjing, China). The specific primers of miRNA were synthesized using the primer design software miRNA Design V1.01, which was provided with the kit, using the mature sequence of miRNA. The reverse transcription reaction of genes was processed using the *Evo M-MLV* reverse transcription kit II (Accuratebiology, Hangzhou, China) in a 20 μL volume system containing 250 ng RNA following the kit instructions.

Except for miRNA, specific primers used for qRT-PCR experiments were designed using NCBI’s online program Primer-blast (www.ncbi.nlm.nih.gov/tools/primer-blast/, accessed on 20 March 2023). All primer sequences are shown in Table 1 and Table 2. All the qRT-PCR reactions were conducted using the SYBR^®^ Green Premix Pro Taq HS qPCR Kit (Accurate biology, Hangzhou, China). *GAPDH* was used as a reference gene to calculate the expression level of target genes using the 2^−ΔΔCt^ method [23]. The reaction conditions of two-step qRT-PCR were as follows: 95 °C for 30 s, 95 °C for 5 s, 60 °C for 30 s, followed by a disassociation stage (default parameters of the instrument). The qRT-PCR amplification system was prepared in 10 µL volumes containing 5 µL of 2 × SYBR^®^ Green Pro Taq HS Premix, 0.2 µL of each primer, 1 µL of cDNA, 0.2 μL ROX reference dye, and 3.4 μL ddH_2_O. Each sample was subjected to three technical replicates. Negative controls with RNase-free water instead of cDNA were included.

## 3. Results

### 3.1. Differentially Expressed Genes and Gene Function Enrichment Analysis of RNA-Seq

Compared with the uninduced petiole (P0, 0 days), bulbil induction development was divided into four periods according to the morphological characteristics of induced bulbil development, namely P1 (3 days), P2 (6 days), P3 (9 days), and P4 (12 days). Compared to the control group, slight enlargement of the petiole tip and formation of growth points were observed on day 6. Four samples from two tissues produced an average of 43.9 million valid reads (ranging from 42.34 to 45.23 million valid reads). There were a total of 8602 significant DEGs (*p* < 0.01, FDR < 0.05) detected between TCb and TCp, among which 4986 were highly expressed in the TCb and 3616 were highly expressed in the petiole.

Gene Ontology (GO) annotation of the 8602 DEGs between TCb and TCp indicated that these DEGs could be categorized into three main classifications: cellular component, molecular function, and biological process (Appendix A). Hypergeometric tests were used in GO enrichment analyses to determine the biological function of these over-enriched DEGs. The enrichment analysis results revealed that the microtubule, tubule complex, microtubule movement, and plasma membrane-related GO terms were highly enriched (Figure 2).

Subsequently, all the DEGs between TCp and TCb were mapped to the KEGG database (Figure 3) with the pathways involving starch and sucrose metabolism (108 DEGs), plant hormone signal transduction (104 DEGs), and phenylpropanoid biosynthesis (76 DEGs) having the highest number of DEGs. The pathway with the highest enrichment degree (rich factor) was for isoflavonoid biosynthesis, followed by indole alkaloid biosynthesis and prodigiosin biosyntheses.

### 3.2. miRNAs Differentially Expressed in TCp and TCb Samples

Since the morphogenetic process of plant organs is regulated by miRNA, we sequenced miRNA expression in TCb and TCp on the 6th day, at which the induced bulbil can be observed. After removing adaptors and low-quality reads, we obtained 4.96 million (TCp1), 7.53 million (TCp2), 5.08 million (TCb1), and 4.35 million (TCb2) valid reads. A total of 171 miRNAs were identified in TCp and TCb, including 118 known miRNAs and 53 novel miRNAs. The expression level was significantly different between TCp and TCb (*p* < 0.05). There were fifteen miRNAs (Appendix A), among which nine miRNAs were highly expressed in TCp.

### 3.3. miRNA Target Gene Prediction and Pathway Enrichment Analysis

We performed target gene prediction and functional enrichment analysis for miRNAs whose expression levels were among the top 15 with significant differences between TCp and TCb. A total of 624 target genes were predicted from 15 miRNAs. Subsequently, KEGG pathway enrichment analysis of the target genes showed that plant hormone signal transduction, mRNA surveillance pathway, and peroxisome pathways were significantly enriched (Figure 4).

### 3.4. Validation of Differential Gene and miRNA Expression by qRT-PCR

qRT-PCR was used for the expression profiles of three miRNAs, their target genes, and three key genes during TCp and TCb at four stages of bulbil induction (Figure 5). Regular differences occurred in the gene expression levels between petiole and bulbil genesis sites in the phytohormone-induced explants. For example, miR167a expression was significantly up-regulated relative to the petiole, reaching a peak in P3 and then plummeting while continuously increasing in the bulbil. The trend of the miR167a target gene, auxin response factor 6 (*ARF6*), in the bulbil and petiole, was opposite to that of miR167a, and completely consistent with auxin-responsive protein 4 (*IAA4*). miR171a and its target gene (*SCL6*) revealed similar spatial-temporal expression characteristics. However, the expression level of *SCL6* was down-regulated relative to the P0 stage, contrary to the results for miR171a. The expression level of miR156a was significantly up-regulated, but the target gene *SPL6* was down-regulated. The expression trend of the *SUS2* gene significantly differed between TCp and TCb, and the expression level of *AMY2* tended to be consistent between tissues and controls.

## 4. Discussion

In recent years, the classification and origin of plant bulbils and the influencing factors, molecular mechanisms, and cytology of bulbils have attracted scholarly attention [24,25,26,27,28,29]. As a plant’s vegetative viviparous organ or special asexual reproductive organ, bulbils have many advantages, such as strong vitality, high survival rate, and short germination time, and are of great significance to plant growth and reproduction and environmental adaptation [30,31].

The bulbil-initiating cells of plants that can produce bulbils in nature originate from the meristem at growth sites, such as in *P. ternata*, *P. cordata*, *Dioscorea polystachya*, and *Lilium sulphureum* [32,33,34,35]. In the research on bulbil induction, an induction system has been established in *L. lancifolium*, *P. ternata*, and *D. polystachya* [11,36,37]. With its development, high-throughput sequencing technology provides important technical support for screening key genes and pathways in the regulation of bulbil formation and development. In this study, we used high-throughput sequencing technology to conduct mRNA and miRNA sequencing analysis on the early bulbil and petiole of *P. ternata* induced by phytohormones; we also screened and identified numerous DEGs between TCp and TCb, and conducted qRT-PCR to verify the expression profiles of miRNAs and their target genes in the four periods of induction. These results provide substantial information for the natural and induced occurrence of *P. ternata* bulbils and the involvement of miRNA in the regulation of bulbil formation.

In the results of the GO enrichment analysis of DEGs, the expression levels of genes in the GO term that were significantly enriched, such as the microtubule, tubulin complex, and microtubule-based biological processes, were significantly higher in TCb than in TCp. Bulbil induction is involved in the dedifferentiation and redifferentiation of plant cells [38], and the dynamic changes in the microtubule and microtubule-related proteins are the key links to ensure cell division and differentiation, growth and development, and cell morphogenesis [39].

In the analysis of KEGG pathway enrichment of DEGs, the notable pathways mainly include starch and sucrose metabolism, plant hormone signal transduction, and phenylpropanoid biosynthesis. Starch and sucrose metabolism pathways enriched 108 DEGs, most of which were highly expressed in bulbils. On the one hand, the bulbil, as a unique vegetative reproductive organ, involved sugar substances, which showed a constant increasing trend with the occurrence and expansion of the bulbil, and ended in the maturity or passive maturity of the maternal wilt [25]. Sucrose is a substrate for transferring and forming starch in the bulbil. Studies on the synthesis and metabolic balance of starch and sucrose during bulbil development revealed that the starch content increased gradually with the development of the bulbil. The metabolism of sucrose was active in the early initiation of the bulbil, and the content of sucrose decreased sharply before the initiation of the bulbil primordium. Moreover, the sucrose and starch content changed with the same trend during the expansion of the bulbil primordium [40]. By contrast, sucrose and other substances may play the role of signaling molecules involved in regulating the bulbil’s initiation and occurrence [25]. Analysis of the expression profiles of a base of Sucrose synthase 2 (*SUS2*) from two tissues during the four stages of induced bulbil development using qRT-PCR indicated that the *SUS2* expression level in TCb decreased first and then increased compared with that in the uninduced stalk; moreover, these outcomes are completely opposite to the trend for the petiole (Figure 5). Evidence suggests that *SUS* may play a role in the development of the shoot apical meristem (SAM) [41]. Further, *SUS* overexpression can significantly improve the growth rate of plants [42,43], and the increase in *SUS* activity in the meristem is postulated to contribute to the increase in cell proliferation. In this study, it was found that the plant hormone signal transduction pathway was one of the more significant pathways in the enrichment results of the KEGG pathways for the target genes of DEGs and DEmiRNAs (Figure 3 and Figure 4).

Plant hormones are involved in the regulation of most life activities in the plant life cycle, and miRNA has become a key regulatory factor in plant hormone response pathways by affecting plant metabolism, distribution, and perception [44]. For example, auxin plays a key role in the entire life process of a plant, and its function is mainly mediated by *ARF* and *AUX*/*IAA* [45]. miR160 and miR167 are dynamic components of the *AUX* response pathway that target the expression of the *auxin response factor* (*ARF*) genes, such as *ARF10*/*16*/*17* and *ARF6*/*8*, to participate in adventitious root initiation, lateral root development, male and female organ maturation, germination, and postembryonic development [46,47,48]. The direct homology of *AtARF* in barley has also been shown to be regulated by miR167 [49].

Studies of somatic embryogenesis (SE) have shown that SE is regulated by miRNA via the plant hormone signaling pathway [50]. In this experiment, the miR167a expression level did not change significantly at the initial stage of bulbil induction and then showed significant histological and temporal differences. This finding is consistent with the early stage of SE, during which miR167 expression levels decreased significantly in the initial dedifferentiation stage and increased in the late dedifferentiation stage [51,52]. miR167 was highly expressed in the late stage of SE of larch and longan [53,54], and the reduction in miR167 activity promoted the formation of corpus callosum and somatic embryogenesis [55]. The expression level of the miR167 target gene *ARF6* was significantly up-regulated in TCp at the initial stage of induction, gradually decreased and maintained equilibrium at P3 and P4 (Figure 1), up-regulated in P2 of TCb, and then decreased, which was similar to the temporal and spatial expression characteristics of auxin-responsive protein 4 (*IAA4*) (Figure 5). *ARF6* and *ARF8* are required for the gradient reaction of auxin, which is essential for somatic embryogenesis, and miR167 affects auxin synthesis and local transport by targeting *ARF6* and *ARF8* in embryogenic callus to regulate plant SE processes [56]. miR171 has been shown to regulate shoot meristem activity and phase transition by repressing *HAIRY MERISTEM* (*HAM*) family genes, which are also named the *LOST MERISTEM* (*LOM*) and *SCARECROW-LIKE* (*SCL*) genes in various plant species [57,58,59,60]. miR171 precursor and mature miR171 species are widely accumulated in different tissues of *Arabidopsis thaliana*, but the MIR171A gene is highly specifically expressed in the epidermis of embryos, vegetative SAM, young leaves and stems, inflorescence SAM, and floral meristem [61,62,63]. The promoter activity of the other three miR171/170 family genes (MIR171B/C, MIR170) was also specifically switched on in the epidermis of shoot meristems at both vegetative and reproductive stages [62]. miR171 participated in the regulation of SE through repression of the Scarecrow-like protein 6 (*SCL6*) gene and played an important role in the regulation of SE at different stages [64,65]. In this study, miR171a was significantly up-regulated in the early stage of bulbil induction (Figure 5) (which was consistent with the early SE accumulation of miR171 in radish (*Raphanus sativus* L.) [66], indicating that miR171a and its target genes played a key role in bulbil induction. However, the similar organizational and temporal expression characteristics and molecular mechanism of miR171a and *SCL6* during bulbil induction require further experimental verification and interpretation (Figure 5). The miR156-*SPL* module plays a crucial role in the initial stage of SE induction [67]. Overexpression of csi-miR156a or inhibition of one of the two target genes, *CsSPL3* and *CsSPL14*, can enhance the SE ability of citrus callus [68]. In the bulbil induction process of *P. ternata* (Figure 5), significant up-regulation of miR156a and significant inhibition of its target gene *SPL6* indicated the important role of the miR156-*SPL* module. However, the specific regulatory effects and related molecular mechanisms of the module still need to be explored further. In the following studies, the physiological and biochemical characteristics, microstructure, regulation of key genes, and molecular mechanism of the *P. ternata* bulbil induction process will be the focus of exploration; this approach will be more conducive to the in-depth investigation of the genesis and development mechanism of *P. ternata* bulbils and the important role of this special plant organ. In addition, *P. ternata* has a simple structure, fast growth rate, relatively fixed bulbil position, relatively stable number of bulbils, and obvious and simple bulbil structure. Thus, *P. ternata* is an excellent candidate for studying plant organogenesis and development and is worthy of further use and research.

## Figures and Tables

**Figure 1 genes-14-01727-f001:**
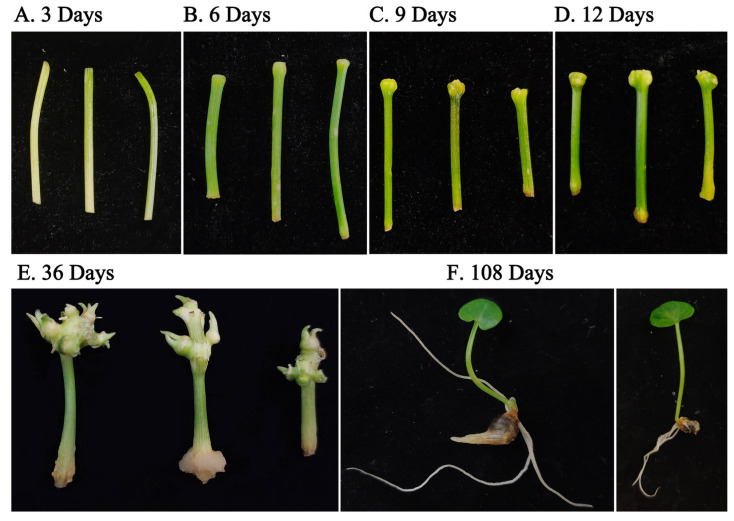
The process of bulbil induction in *P. ternata*. 3 days (**A**), 6 days (**B**), 9 days (**C**), and 12 days (**D**) show the early stage of bulbil induction, and 36 days (**E**) and 108 days (**F**) show the mature induced bulbil.

**Figure 2 genes-14-01727-f002:**
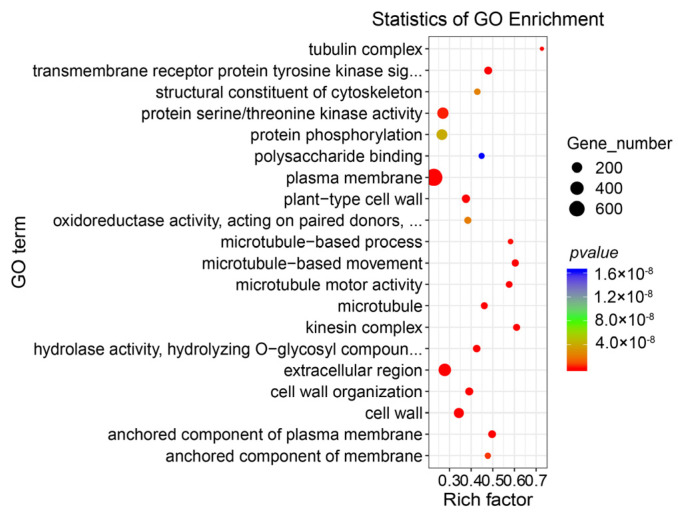
The GO enrichment analysis of TCb and TCp differentially expressed genes (DEGs). The top 20 GO terms with significant enrichment; rich factor indicates the enrichment degree of differential genes, and the higher the value, the higher the enrichment degree.

**Figure 3 genes-14-01727-f003:**
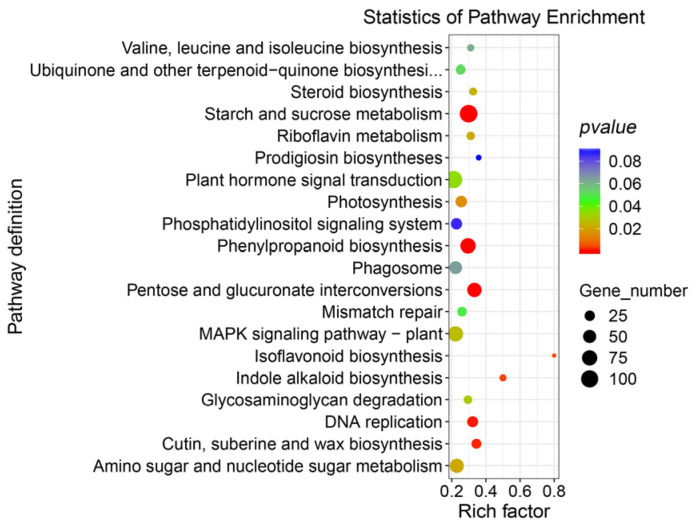
The KEGG pathway enrichment analysis of TCb and TCp differentially expressed genes (DEGs). The top 20 KEGG pathways with significant enrichment; rich factor indicates the enrichment degree of differential genes, and the higher the value, the higher the enrichment degree.

**Figure 4 genes-14-01727-f004:**
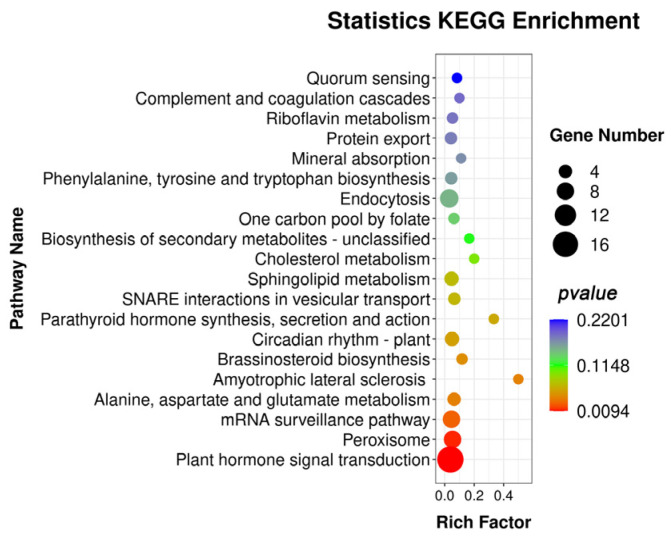
The KEGG pathways enrichment analysis target genes of differentially expressed miRNAs (DEmiRNAs). The top 20 KEGG pathways with significant enrichment; rich factor indicates the enrichment degree of DEmiRNAs, and the higher the value, the higher the enrichment degree.

**Figure 5 genes-14-01727-f005:**
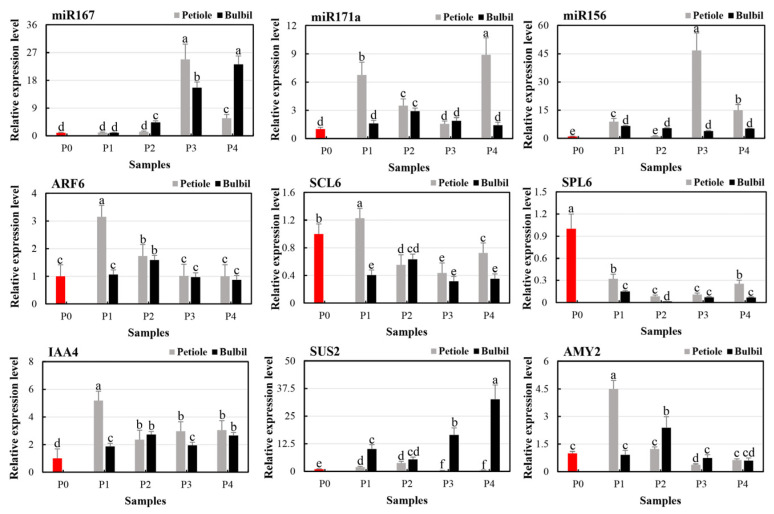
Verification of expression profiles of key genes, miRNAs, and their target genes via qRT-PCR. P0 (red) was an uninduced petiole and served as a blank control, P1 = 3 d, P2 = 6 d, P3 = 9 d, P4 = 12 d. There was no significant difference with the same letter (*p* > 0.05), while there was a significant difference without the same letter (*p* < 0.05).

**Table 1 genes-14-01727-t001:** Detailed information and primers of miRNA.

Gene ID	miRNA Sequences	Reverse Transcription Primer	Primer Sequences (Forward/Reverse, 5′–3′)
aly-miR167a-5p	TGAAGCTGCCAGCATGATCTGA	GTCGTATCCAGTGCAGGGTCCGAGGTATTCGCACTGGATACGACTCAGAT	CGTGAAGCTGCCAGCATG AGTGCAGGGTCCGAGGTATT
mes-miR171a	TTGAGCCGCGTCAATATCTCC	GTCGTATCCAGTGCAGGGTCCGAGGTATTCGCACTGGATACGACGGAGAT	CGTTGAGCCGCGTCAAT AGTGCAGGGTCCGAGGTATT
sbi-miR156a_L 1	TGACAGAAGAGAGTGAGCAC	GTCGTATCCAGTGCAGGGTCCGAGGTATTCGCACTGGATACGACGTGCTC	GCGCGCTGACAGAAGAGAGT AGTGCAGGGTCCGAGGTATT

**Table 2 genes-14-01727-t002:** Detailed information and primers of key genes and miRNA target genes.

Gene	Annotation	Homolog Locus	Primer Sequences (Forward/Reverse, 5′–3′)	Product Length
*ARF6*	Auxin response factor 6-like isoform x1	XM_026023239.1	AGGGCGATGTTCTTCTCGTC GGGTCTTGATGGCTCGCATA	187 bp
*IAA4*	Auxin-responsive protein iaa4 isoform x3	XM_015774120	ATCTGAGGCTGGGGTTTAGC TCACGAAGAAGGTTGCTTGC	192 bp
*SCL6*	Scarecrow-like protein 6	XM_015771585	GTTCCAGTAGCACACCTCCC CCACCAAACCCGATGTCGAA	184 bp
*SPL16*	Squamosa promoter-binding-like protein 16	XM_015793891.2	TGATCGAGGAAATGAGGCCG CGAACTTGAGGTAGGGGCAG	189 bp
*SUS2*	Sucrose synthase 2	XM_039929826.1	GCGGAGATCATAGTGGACGG ACAGCGTCATCAACCTCTCTG	199 bp
*AMY2*	α amylase	XM_015779949.2	AAGGAGAGCTATGGCGACTG GTGGGTATTCCTGGGTGTGT	187 bp

## Data Availability

All RNA-seq induced bulbil formations are openly available in the National Center Biotechnology Information Sequence Read Archive (SRA) database with accession numbers of SRR25252584 (*P. ternata*) and SRR25252583 (*P. ternata*).

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
