# Peer review of "mRNA-Seq and miRNA-Seq Analyses Provide Insights into the Mechanism of Pinellia ternata Bulbil Initiation Induced by Phytohormones"

_genes, 2023, doi:10.3390/genes14091727_

Round 1

Reviewer 1 Report

This study by Xu et al. carried out the transcriptome sequencing of both mRNA and miRNA in the induced petiole (TCp) and the induced bulbil (TCb), and aimed to provide useful clues for the identified bulbil development-related miRNAs and targets. However, there are obviously morphology differences between the induced petiole (TCp) and the induced bulbil (TCb). It should be obtaining lots of DEGs if do the transcriptome sequencing of either mRNA or miRNA. The authors need to polish more which genes are bulbil development-related miRNAs and targets.

The major comments listed below

1) More analysis or experiment needs to be added. The genes induced by 6-BA and NAA treatment are a quite classic/old story, it will lose the novelty if couldn’t polish or verify which candidates or a set of genes/miRNAs are important for bulbil development-related miRNAs and targets.

2) The figures’ quality looks very poor. It’s unclear, especially the text labeling. It needs to be improved.

Other minors:

1)     Line 163, “the DEGs between TCb and TCb were mapped to the KEGG database”, there should be “TCp and TCb”.

2)     There should be a space between number and unit, such as “3cm”, “0.5mg/L”, … “20uL", “250ng”, ……

3)     Suggest the author transfer the primers list in Table 1, and 2 to the supplemental table 1.

The paper is fine to read. but there are minor formatting errors to correct.

Author Response

We are grateful to the reviewers for their thorough reading and helpful and insightful comments. These comments and suggestions will greatly assist us in improving the manuscript. We have carefully studied the comments and made corrections in the hope of approval. Revised sections are marked in the paper. The main corrections of the paper and the reply to the reviewer's comments are as follows:

General comments

1) More analysis or experiment needs to be added. The genes induced by 6-BA and NAA treatment are a quite classic/old story, it will lose the novelty if couldn’t polish or verify which candidates or a set of genes/miRNAs are important for bulbil development-related miRNAs and targets.

Responses 1: Thanks to the reviewer for the comments. The opinions you put forward are the ultimate purpose of this series of experiments, which is what we have been exploring and pursuing. In this experiment, the key point that we pay attention to and are interested in is the induction process of P. ternata bulbil formation, which depends on the special organ characteristics of the bulbil, which is different from other plants. With this experiment, we can focus on several important pathways, such as plant hormone signaling and starch and sucrose metabolism, including the regulatory role of miRNAs in these biological processes. In addition, we also conducted transcriptome sequencing comparisons between the bulbil and petiole of P. ternate under normal growth conditions. The obtained experimental results are similar to those between the induced bulbil and petiole (Unpublished data). Based on the sequencing and expression analysis results, we have preliminarily identified miR156, miR167, and miR171, which may exhibit specific regulatory roles during bulbil development. We are currently conducting more in-depth and detailed research and investigation into these findings. The relevant research results will be published in another manuscript. These finding will provide some clues for the molecular mechanism of P. ternata bulbil formation and development.

2) The figures’ quality looks very poor. It’s unclear, especially the text labeling. It needs to be improved.

Responses 2: Thanks a lot for the reviewer's suggestion. We have improved the clarity of all figures and replaced the ones affecting readability (Figure 2 and Figure 4) to present the results more accurately and specifically.

Others:

1) Line 163, “the DEGs between TCb and TCb were mapped to the KEGG database”, there should be “TCp and TCb”.

Responses 1: We greatly appreciate the careful review by the reviewers. We have corrected this mistake and reexamined and rectified other errors in the manuscript. Once again, we thank you for your attentiveness and professionalism.

2) There should be a space between number and unit, such as “3 cm”, “0.5 mg/L”, … “20 uL", “250 ng”, ……

Responses 2: We appreciate the recommendation made by the reviewer. We have corrected all the format issues.

3) Suggest the author transfer the primers list in Table 1, and 2 to the supplemental table 1.

Responses 3: Thanks a lot for the reviewer's suggestion. We have reformatted Table1 and 2 to ensure they are easy to read and in the right place. However, after many discussions and careful consideration, table information is an essential part of the main body of the paper, which is more conducive to readers' intuitive reading and analysis of the article content and relevant conclusions. However, thank you for your suggestions, which will improve us in the future.

Reviewer 2 Report

The manuscript titled "Insights into the Mechanism of Pinellia Ternata Bulbil Initiation Induced by Phytohormones through mRNA-seq and miRNA-seq Analyses" presents a comprehensive screening and identification of numerous differentially expressed genes (DEGs) between bulbil initiation stages. Moreover, qRT-PCR was conducted to verify the expression profiles of miRNAs and their target genes in the four induction periods. Consequently, this study offers significant information regarding both the natural and induced occurrences of P. ternata bulbil and the role of miRNA in regulating bulbil formation. However, it is worth noting that only two biological replicates were utilized for each RNA-seq sample, and each biological replicate consisted of at least five petioles or bulbil. Hence, it would be valuable to ascertain whether these two biological replicates were randomly selected and how many replicates were grown in vitro. These numerical values offer an indication of the level of reproducibility present within the obtained results.

Introduction, It is highly desirable to provide a comprehensive depiction of the abbreviations of each growth regulators, particularly upon their initial appearance.

Other details:

Line 43, add a space between the number and mg in:   0.5 mg/L

Lines 144 to 147, Table 1 is mixed in-between the Result text.

Figures 2, 3, 4 and 5, the quality must be improved

Author Response

We are grateful to the reviewers for their thorough reading and helpful and insightful comments. These comments and suggestions will greatly assist us in improving the manuscript. We have carefully studied the comments and made corrections in the hope of approval. Revised sections are marked in the paper. The main corrections of the paper and the reply to the reviewer's comments are as follows:

General comments

1) However, it is worth noting that only two biological replicates were utilized for each RNA-seg sample, and each biological replicate consisted of at least five petioles or bulbil, Hence, it would be valuable to ascertain whether these two biological replicates were randomly selected and how many replicates were grown in vitro.

Responses 1: The suggestion of the reviewer is much appreciated. Taking into consideration experimental costs, we chosed two biological replicate samples for RNA-seq. Before sequencing, the five biological replicates for each time point (each replicate composed of 10-15 individual samples) were pooled into two samples to ensure reproducibility of experimental results. We have added the relevant explanation in "2.1 Plant materials" to aid in understanding the experimental design.

2) Introduction, lt is highly desirable to provide a comprehensive depiction of the abbreviations of each growth regulators, particularly upon their initial appearance.

Responses 2: Thanks a lot for the reviewer's suggestion. We have already added the corresponding comprehensive depiction of each growth regulators in the introduction.

Others:

1) Line 43, add a space between the number and mg in:0.5 mg/L.

Responses: We are very grateful to the reviewer for his carefulness. We have correctly added spaces at 0.5 mg/L.

2) Lines 144 to 147.Table 1 is mixed in-between the Result text.

Responses: We appreciate the reviewer's suggestion. We have adjusted the table to ensure that it is in the most appropriate position.

3) Figures 2, 3, 4 and 5, the quality must be improved.

Responses: Thanks a lot for the reviewer's comments. We have improved the clarity of all the figures and replaced the ones affecting the readability (Figure 2 and Figure 4) to present the results more accurately and specifically.

Round 2

Reviewer 1 Report

Thanks for the authors's reply and explaination. The authors addressed all my comments and did the revision. 

Just make sure that the figures quality are good enough before publication. In the combined pdf from my end, it's still unclear.